



# Temporal Characteristics and Atmospheric Drivers of Onsets and Terminations of Soil Moisture Droughts in Europe

Woon Mi Kim[1], Santos J. González-Rojí[2,3], Isla R. Simpson[1], and Daniel Kennedy[1,4]

[1]Climate and Global Dynamics Laboratory, NSF National Center for Atmospheric Research, Boulder CO, United States
[2]Climate and Environmental Physics, University of Bern, Bern, Switzerland
[3]Oeschger Centre for Climate Change Research, University of Bern, Bern, Switzerland
[4]Earth Research Institute, University of California, Santa Barbara CA, United States
**Correspondence:** Woon Mi Kim (wmikim@ucar.edu)

**Abstract.**

Many studies have focused on understanding the drivers of soil moisture droughts in Europe when the events have already intensified. Still, how atmospheric circulation changes throughout the entire life cycle of droughts, particularly during the transition periods to drought initiation and termination, has not been thoroughly investigated. Therefore, this study investigates

temporal characteristics and atmospheric circulation associated with onsets (a transition period from a normal condition to drought) and terminations (a transition period from drought to a normal condition) of soil moisture droughts in Europe during 1980–2020. The typical duration, preferred seasons of occurrence, and atmospheric circulation during onsets and terminations are examined. The regions of study are central (CEU) and Mediterranean (MED) Europe, and soil moisture from ERA5-Land, GLEAM version 3, SoMo.ml, Noah-LSM, and a simulation from the Community Land Model version 5 (CLM-TRENDY) are

utilized.

Our findings indicate that the duration of onsets and terminations depends on the dataset: the five soil moisture datasets exhibit different mean duration of onsets and terminations, with ERA5-Land showing shorter and GLEAM longer duration across Europe. Nevertheless, within the same dataset, onsets and terminations exhibit similar durations, implying that onsets and terminations can occur at the same speed. Regarding the preferred seasons for onsets and terminations, onsets occur

more during the wet seasons, specifically summer in CEU and autumn and winter in MED. Nevertheless, the frequencies of occurrence during these seasons only slightly exceed that during other seasons. Terminations tend to occur more during the driest seasons. For atmospheric circulation, onsets come with large-scale anticyclonic atmospheric circulation patterns. Terminations do not exhibit a dominant mean circulation pattern over the region where terminations occur but instead show geopotential height anomalies with reduced magnitudes in Europe and a high-pressure system over the North Atlantic. The

anticyclonic circulation during onsets is anomalously persistent and shows linkages with the North Atlantic Oscillation (NAO). Positive NAO occurs much more frequently during onsets compared to other drought phases. This finding implies an important role for this large-scale mode of variability in initiating dry periods by reducing soil moisture and its potential to serve as an early warning for droughts during the period prior to the events.



## 1 Introduction

Drought is a climate phenomenon that is characterized by a prolonged period of dry conditions (Dai, 2011). When depletion of water occurs in the soil, the event is referred to as a soil moisture drought. Soil moisture droughts can result from a persistent dry period with little precipitation (meteorological drought) alone or can arise from the combined influence of reduced precipitation and increased evapotranspiration. The impacts of soil moisture droughts can be far-reaching, affecting ecosystems and a wide range of socioeconomic sectors (Naumann et al., 2015; Bachmair et al., 2016) with clearly apparent impacts on

agricultural activities (Moravec et al., 2021). The damage caused by droughts becomes increasingly severe with the increasing duration of the events. Although dry-wet fluctuations are a natural aspect of a region's hydroclimate, there is evidence that anthropogenic warming has perturbed the natural characteristics of droughts through an increase in vapor pressure deficit and the intensification of the global hydrological cycle (Trenberth, 2011; Van Loon et al., 2016; Seneviratne et al., 2021).

Determining the precise timing of initiation and termination of droughts is challenging, primarily due to the complex and

multifaceted nature of droughts (Cook et al., 2018). The drivers of a drought can be diverse, from large-scale climate patterns to regional-scale processes, including synoptic weather systems and land-atmosphere interactions. Each of these processes has different time scales of influence on soil moisture variability. Another complexity arises from the fact that droughts are typically slow events, whose impacts on the ecosystem become noticeable when the events have already reached the mature phase, with some accumulated periods of precipitation or soil moisture deficit (Wilhite, 2000). These factors collectively make

it challenging to define when exactly a dry or wet condition that initiated or ended a drought has commenced. However, investigating drought phases, particularly periods of transitions to drought onsets and terminations, is a topic that requires more attention to improve early predictions of such events and prepare suitable mitigation strategies.

Several studies that investigated the life cycle of droughts have defined drought phases based on different standardized drought indices, accumulated precipitation deficit, or soil moisture anomalies (e.g., Mo, 2011; Seager et al., 2019; Shah and

Mishra, 2020; Řehoř et al., 2021). Although the criteria to divide the life cycle of a drought into different phases vary across the studies, in general, a drought initiation is the period preceding a drought threshold, and a drought termination is the recovery period from the drought threshold or from the minimum soil moisture or precipitation deficit to normal conditions. The typical duration of drought phases and the drivers involved in these different phases of the drought life cycle have been investigated in these studies.

Focusing on the United States, Mo (2011) examined the duration and large-scale drivers associated with onsets and terminations of meteorological and soil moisture droughts on various time scales across the United States using the North American Land Data Assimilation System (NLDAS-1) dataset during 1916–2007. This study shows that drought onsets are slower than terminations, as onsets require a certain accumulation period of precipitation deficits until they can progress into droughts, whereas heavy precipitation events can more easily alleviate drought. Due to a longer development period, onsets are expected

to be more predictable than terminations. In contrast, using the models from NLDAS-2, Seager et al. (2019) showed that over the southern Great Plains, there is no difference in the duration between onsets and terminations of soil moisture droughts on a seasonal time scale. They suggest that this mismatch in the duration between these two studies may arise from the use of



different definitions of onsets and terminations. Mo (2011) defined onsets (or the transition periods to onsets) as all preceding
months with a precipitation deficit in the year prior to reaching the drought threshold. Seager et al. (2019) considered onsets
as transition periods from normal (above a negative unit standard deviation of soil moisture anomalies) to below the drought
threshold (below a negative unit standard deviation of soil moisture anomalies, with the change over the season greater than
one standard deviation). Still, both studies found that large-scale atmosphere-ocean climate anomalies in the tropics are the
drivers of onsets and terminations of droughts.

For other regions, Shah and Mishra (2020) investigated characteristics of onsets and terminations of different types of
droughts in India based on the soil moisture and surface datasets from the Variable Infiltration Capacity (VIC) model. Onsets
and terminations of droughts in India tend to occur more frequently during the summer monsoon seasons, and their long-term
variability is linked with El Niño Southern Oscillations (ENSO) and Indian Ocean Dipole (IOD). For Europe, Řehoř et al.
(2021) examined atmospheric circulation anomalies associated with the three phases of soil moisture droughts in the Czech
and Slovak Republics during 1961–2019. Using the output from the SoilClim model, they found that anticyclonic circulation
types linked with low precipitation occur more frequently during the initiation (onsets) and throughout droughts. They found
that the opposite cyclonic circulation types that bring precipitation to the region were more frequently observed during the
recovery phase of droughts. Focusing on the British Isles, Parry et al. (2016) reviewed drought terminations in different studies,
describing the various synoptic phenomena that end drought events in Europe.

In the last decade, Europe has experienced multiple intense droughts, marked by low soil moisture and accompanied by
record-breaking summer heatwaves (e.g., Hari et al., 2020; Sousa et al., 2020; Moravec et al., 2021; Rakovec et al., 2022).
Various studies have examined the drivers of individual droughts during this period. For example, the intense summer 2015
(Ionita et al., 2017) and 2018 (Moravec et al., 2021) droughts were driven by upper-level ridges and blocking highs, which
were associated with a weakening of the subtropical jets. The 2016/2017 drought that affected almost the entire Europe was
characterized by consecutive blocking events and subtropical ridges, which in turn weakened zonal circulation and moisture
transport from the Atlantic (García-Herrera et al., 2019). During this drought event, the impact of temperature that drove an
increase in atmospheric evaporative demand was more pronounced in southern Europe, and a similar situation was found for
the 2022 event (Faranda et al., 2023). Vicente-Serrano et al. (2021) have indicated that the contribution of vapor pressure deficit
on droughts has significantly increased in western Europe since the mid-20th century.

Despite these significant advances in the understanding of soil moisture droughts and drivers of the individual events
presented above, the climatological characteristics of drought onsets and terminations in Europe are still a relatively under-
explored topic. Climatological analyses of droughts have been conducted using standardized drought indices (e.g., Lloyd-
Hughes and Saunders, 2002; Spinoni et al., 2015), but they did not focus on the characteristics of each of the drought phases.
Thus, there is a need to improve understanding of the temporal characteristics and the atmospheric circulation patterns in-
volved in the different phases of soil moisture droughts in Europe, particularly during onsets and terminations, to enhance
early predictions and readiness for such extreme events.

The objective of this study is to investigate temporal characteristics of onsets and terminations and typical circulation patterns
of soil moisture droughts in Europe during 1980–2020. First, we examine the climatological duration and preferred seasons of





occurrence of drought onsets and terminations. We also assess the relationship between the duration of onsets, terminations, and droughts, aiming to understand whether long (short) droughts require long (short) transition periods to begin or to recover

from. Aspects of the surface water balance (precipitation, precipitation minus evaporation, and soil moisture) involved in this relationship are also assessed.

Second, we analyze typical atmospheric circulation patterns of soil moisture drought onsets and terminations. One of the aims is to understand atmospheric processes that help onsets progress into droughts in CEU and MED. Then, particular focus is given to the association of different drought phases with the North Atlantic Oscillation (NAO) to address the large-scale

influence on drought phases and whether NAO serves as an early warning for soil moisture droughts in Europe.

The study regions are central (CEU) and Mediterranean Europe (MED), confined approximately between 45°N–56.5°N and 9.5°W–25.5°E for CEU and between 36.5°N–45°N and 9.5°W–25.5°E for MED following the division based on the IPCC climate reference regions (Iturbide et al., 2020). These are the regions in Europe that have been most affected by droughts in the last decade and where soil dryness is expected to increase in future warming scenarios, with medium (in central Europe)

and high (in southern Europe) confidence levels (Seneviratne et al., 2021).

The typical duration and seasonality of onsets and terminations are estimated using five soil moisture datasets. As most of the previous literature has employed only one soil moisture dataset or datasets from a single project, by using several soil moisture datasets from diverse projects, we aim to examine the robustness of the temporal characteristics of drought. The description of the datasets is given in Section 2. The definition of drought phases and other methods employed in the analysis

are introduced in Section 3. All the results on the climatological duration, seasonality of onsets and terminations, and typical circulation patterns are provided in Section 4. Finally, we present the discussion and conclusion in Section 5.

## 2 Data

Our study regions are central (CEU) and Mediterranean Europe (MED), extended approximately between 45°N–56.5°N and 9.5°W–25.5°E for CEU and between 36.5°N–45°N and 9.5°W–25.5°E for MED (Fig. 1c) based on the division following on

the IPCC climate reference regions (Iturbide et al., 2020). The analyzed period is 1980–2020.

The atmospheric variables we use are monthly averaged geopotential height at 500 hPa and precipitation obtained from the ERA5 (Hersbach et al., 2020) reanalysis. ERA5 is the latest reanalysis product of the European Centre for Medium-Range Weather Forecasts (ECMWF) and is generated with the 2016 version of the ECMWF numerical weather prediction model and the integrated forecasting system Cy41r2 data assimilation. The atmospheric variables have a horizontal resolution of 0.25° ×

0.25°, covering the period from 1940 until the present.

As an alternative precipitation dataset, we use E-OBS (version 27.0e;  Cornes et al., 2018), which provides gridded daily precipitation amounts over Europe (land-only, 25°N–71.5°N, 25°W–45°E). E-OBS is based on weather station data collected by the ECA&D initiative (Klein Tank et al., 2002; Klok and Klein Tank, 2009), and the final precipitation values are generated as the means of an ensemble with 100 members of the daily precipitation estimates (Cornes et al., 2018). The spatial resolution

of E-OBS is 0.1° × 0.1°, and the data are available from 1950 onward.



We take Hurrell's station-based North Atlantic Oscillation Index (NAO) obtained from the NCAR climate data guide (Hurrell et al., 2023). The index represents the fluctuation between the Icelandic low and the Azores high, calculated as the difference of normalized sea level pressure (SLP) between the stations in Lisbon, Portugal and Stykkisholmur/Reykjavik, Iceland (Hurrell et al., 2003).

For soil moisture, we use five gridded products: three are the output of land surface models (LSM) forced by observation-based meteorological fields, one is based on a multi-layer soil model and assimilated with satellite-based products, and the last one is a machine-learning-trained observation-based dataset. The three soil moisture datasets from LSMs are ERA5-Land (Muñoz-Sabater et al., 2021), Noah-LSM (Koren et al., 1999) from the Global Land Data Assimilation System project (GLDAS; Rodell et al., 2004), and TRENDY (Friedlingstein et al., 2022) from the Community Land Model version 5 (CLM5; Lawrence et al., 2019). The next dataset is Global Land Evaporation Amsterdam Model (GLEAM; Miralles et al., 2011; Martens et al., 2017) soil moisture, and the last one is SoMo.ml, which is derived from a machine learning-based model (O and Orth, 2021). Where available, evapotranspiration is also retrieved from these datasets for analysis.

We mostly rely on outputs from gridded datasets because our study requires continuous soil moisture data to identify drought phases properly, and many solely observation-based soil moisture datasets are generally short and not continuous in time and space. The datasets are summarized in Table 1, and a brief description of each dataset follows.

ERA5-Land (Muñoz-Sabater et al., 2021) uses the offline ERA5 Land surface model, and it is forced by the atmospheric variables from the ERA5 reanalysis. The land processes are based on the ECMWF Scheme for Surface Exchanges over Land with land surface hydrology from the H-TESSEL model. The horizontal resolution is $0.25° \times 0.25°$, and we employ the data with the monthly time resolution.

GLEAM version 3 (Miralles et al., 2011; Martens et al., 2017) is a set of algorithms to estimate global evaporation that aims to provide an advanced representation of evaporation based on satellite and reanalysis forcing. The dataset contains not only terrestrial evaporation but also surface and root-zone soil moisture. Soil moisture is estimated using a multi-layer running-water balance model, and the upper-level soil moisture is assimilated with satellite-based microwave soil moisture. The spatial resolution of the dataset is $0.25° \times 0.25°$ covering the period 1980 to the present.

SoMo.ml v1 (SoMo; O and Orth, 2021) is a global daily soil moisture dataset reconstructed through a machine learning model trained with in-situ soil moisture measurements across the globe. SoMo employs a Long Short-Term Memory neural network to reconstruct the daily global soil moisture field. The predictors fed into the model are the meteorological variables from reanalysis and remote sensing datasets, and the target variable is soil moisture from 1000 in-situ measurements across the globe. The means and standard deviations of the daily in-situ soil moisture are scaled up to match those of the ERA5 grid cells to produce seamless merging across different stations and time series. The horizontal resolution of SoMo is $0.25°$, and the daily temporal resolution is converted to the monthly values by calculating the monthly averages. The temporal coverage of the dataset is from 2000 to 2019.

GLDAS (Rodell et al., 2004) is a project comprising various land surface models that provide global land surface variables. The output from GLDAS 2.1 is used for the period 2000–2020. The atmospheric input forcing for GLDAS combines data from 160 models and observations, including the Princeton meteorological forcing (Sheffield et al., 2006). From the LSMs that comprise



GLDAS, we use Noah LSM (Noah; Koren et al., 1999) that provides the surface level soil moisture. We employ the dataset with the spatial resolution of 1° × 1° and the monthly average temporal resolution.

TRENDY (CLM-TRENDY) is a simulation from the offline CLM5 (Lawrence et al., 2019) and is part of the Global Carbon Budget 2022 project (Friedlingstein et al., 2022). The simulation was performed with a transient $CO_2$ and land use change from
1701 to 2021, and forced by the merged Climate Research Unit (CRU) – Japanese 55-year Reanalysis (JRA55) atmospheric forcing from 1901 onward. Before 1901, the atmospheric forcing during 1901–1920 is cycled over to fill the period. The dataset has a spatial resolution of approximately 0.95° × 1.25° and the monthly temporal resolution.

**Table 1.** Soil moisture datasets employed in this study.

| Dataset name (Abbreviation) | Institution | Type | Temporal resolution | Horizontal resolution | Reference |
|---|---|---|---|---|---|
| ERA5-Land (ERA5-Land) | ECMWF | Offline LSM | 1950–Present | 0.25° × 0.25° | Muñoz-Sabater et al. (2021) |
| GLEAM v3 (GLEAM) | ESA | Multi-layer soil model and assimilated | 1980–2021 | 0.25° × 0.25° | Miralles et al. (2011) Martens et al. (2017) |
| SoMo.ml (SoMo) | MPI Biogeochemistry | Machine-learning trained model | 2000–2019 | 0.25° × 0.25° | O and Orth (2021) |
| Noah-LSM from GLDAS v2.1 (Noah) | NASA | Offline LSM | 2000–Present | 1° × 1° | Rodell et al. (2004) |
| CLM-TRENDY (CLM-TRENDY) | NCAR | Offline LSM | 1701–2021 | 0.95° × 1.25° | Lawrence et al. (2019) Friedlingstein et al. (2022) |

# 3 Methods

## 3.1 Metrics for soil moisture droughts

We use soil moisture from the surface level, which is 10 cm (SM10). This layer is chosen as it is the most commonly available among the datasets. GLEAM, SoMo, Noah, and CLM-TRENDY directly output 10 cm soil moisture. For ERA5-Land, as this level is not available, the corresponding value is obtained by estimating how much soil moisture is in the first 3 cm of the second layer (7–28 cm), then adding this amount onto the soil moisture of the first layer (7 cm). The assumption is that there is no gradient of soil moisture within the second layer.

Although the surface layer soil moisture cannot indicate dryness occurring in deeper soil layers, this depth is where soil moisture can be measured over a wide spatial extent and is commonly available in most soil moisture products. Moreover, SM10 is highly correlated to the commonly used operational drought indices, for instance, the Standardized Precipitation Evapotranspiration Index (SPEI; Vicente-Serrano et al., 2009). The variable is directly affected by changes in atmospheric circulation; hence, it is suitable for examining contemporaneous influences of atmospheric drivers on soil moisture.



The drought index to quantify soil moisture droughts is the three-month running mean of standardized surface soil moisture anomalies, same as Mo (2011). Running averages over three months smooth out noises in the monthly time series by taking into account the typical time scale of a season in the mid-latitudes. The anomalies are calculated at each grid point over the study regions. To obtain these soil moisture anomalies, first, the time series of soil moisture are deseasonalized through subtraction of the 2000–2014 (15 years) annual cycles, and then they are standardized using the multi-year standard deviations for each month

of the same period. These 15 years are used for standardization as they are common to all five datasets and exclude the intense soil moisture dryness in CEU that began in 2015. The three-month running means are calculated using these standardized soil moisture anomalies (from now on, denoted as $\Delta SM$). These time series have a monthly time resolution, where an individual timestep $t$ encompasses the soil moisture conditions over the previous three cumulative months, from $t_{-2}$ to $t_0$. For instance, the anomaly of February 1981 indicates the soil moisture conditions during December 1980–February 1981. This approach of

defining a drought index with a non-centralized moving average has an operational purpose, making it suitable for monitoring drought conditions for the current timestep. Additionally, this method aligns with other existing drought indices such as the Standardized Precipitation Index (SPI) (McKee et al., 1993) and SPEI.

### 3.2    Definitions of drought phases: onset, drought, and termination

We split the life cycle of drought into three phases: an onset (O), a drought (D), which is an intense dry period after the onset,

and a termination (T; O, D & T to refer to all three drought phases). In our study, onsets and transitions are the transition periods leading to droughts and to normal conditions, respectively. An entire O, D & T is composed of consecutive negative $\Delta SM$ with at least one $\Delta SM$ in timestep $t$ (i.e., a 3-month average) falling below a minus one standard deviation (-1$\sigma$) during the drought. The entire phase then finishes with a positive $\Delta SM$. The detailed definitions of each phase are the following, which are also illustrated in Fig 1a and b:

–   An onset within a drought life cycle (an O, D & T) is a transition period from above zero $\Delta SM$, without including this positive $\Delta SM$, to a drought ($\Delta SM$ below -1$\sigma$). It is composed of the period prior to $\Delta SM$ falling below -1$\sigma$ during which -1$\sigma<\Delta SM \leq 0$. This means that onset is the consecutive period of below normal (below zero) $\Delta SM$ prior to crossing a drought threshold (-1$\sigma$). In situations where a drought begins rapidly, with $\Delta SM$ falling below -1$\sigma$ without any transition period with $-1\sigma<\Delta SM \leq 0$, the onset is said to have a duration of one-month timestep. This definition

ensures a minimum onset period of one month, taking into account the monthly resolution of $\Delta SM$.

–   A termination is defined as the period over which $\Delta SM$ transitions from below -1$\sigma$ to above zero, without including the timestep with a positive $\Delta SM$. It is a transition period from the end of a drought to a normal state. It is counted backward from the first time $\Delta SM$ exceeds zero after a drought event, back to the timestep after the most recent timestep with $\Delta SM < $ -1$\sigma$. Similar to the onset, if there is no transition period with $-1\sigma<\Delta SM \leq 0$, then the duration of termination

is considered to be one month.





- A drought is the rest of the months within a drought life cycle excluding O&T. It commences with the first occurrence of $\Delta SM \leq -1\sigma$ and continues until the last timestep of the life cycle where $\Delta SM \leq -1\sigma$. $\Delta SM$ during this period must be negative but is not necessarily always below $-1\sigma$.

- Additionally, we define a no-drought dry period (NDD). This period is characterized by $-1\sigma < \Delta SM < 0$ without ever
progressing into an actual drought with $\Delta SM \leq -1\sigma$ (Fig. 1b).

A drought threshold of $-1\sigma$ in a standardized drought index defined in this study corresponds approximately to the 15.9th percentile level. Based on the classification of drought categories in other standardized drought indices such as SPI or SPEI (Lloyd-Hughes Benjamin and Saunders Mark A., 2002), values below this threshold indicate moderate to extreme droughts. Our definition of O & T follows a similar approach to that of Seager et al. (2019), with the same threshold of $-1\sigma$. But our
analysis uses the drought metric with a monthly time scale, similarly as Mo (2011).

### 3.3  Temporal characteristics of droughts

We calculate the duration of onsets and terminations by counting the number of timesteps within the onsets or terminations for each drought event at each grid point in the study regions. Statistical assessment to compare the mean duration between onsets and terminations and to compare the duration of onsets and terminations between the datasets is performed using t-tests (Wilks,
2011) within CEU and MED separately. For this, the durations of all individual O&T are collected and separated into CEU and MED. For the t-tests, the null hypothesis assumes that the means of the two phases or two datasets are derived from the same population. Our test, therefore, assesses whether their means are inconsistent with being sampled from the same population at a 95% level using a two-sided test.

The preferred seasons for O&T's are estimated using ERA5-Land, GLEAM, and CLM-TRENDY since they cover a longer
time period (1980–2020). Thus, we can include more drought events in that analysis. Since the study regions are located in the mid-latitudes, a standard division into the four boreal seasons is adopted (DJF is winter, MAM is spring, JJA is summer, and SON is fall). We estimate the preferred seasons of the last month of onset (the timestep before crossing $-1\sigma$) and the first month of termination (the timestep when $\Delta SM$ first crosses above $-1\sigma$ within the termination). An onset (here, the last month of onsets) or termination (the first months of termination) is said to occur in a given season if it falls within that given
season. The number of O&T's within a season is counted at each grid point, and then, the ratio between the number of onset (or termination) occurrences during a given season and the total onset (or termination) occurrences is calculated. If droughts occur uniformly across seasons, the occurrence ratio would be 0.25 for each season.

The preferred seasons for O&T's are also assessed on a continental scale, considering more subdivisions within CEU and MED. This is done to take into account different mean hydroclimate conditions within the region. For this purpose, the two
study regions, CEU and MED, which are divided following Iturbide et al. (2020), are then separated into six, following similar divisions employed by Christensen and Christensen (2007) (Fig. 1c): Iberian Peninsula (IP), east-southern Europe (EMED), Alps (ALP), France (FR), mid-Europe (MCEU), and east-central Europe (ECEU). A median of the occurrence ratio of O&T in a season across the grid points over each subdivision is then calculated.



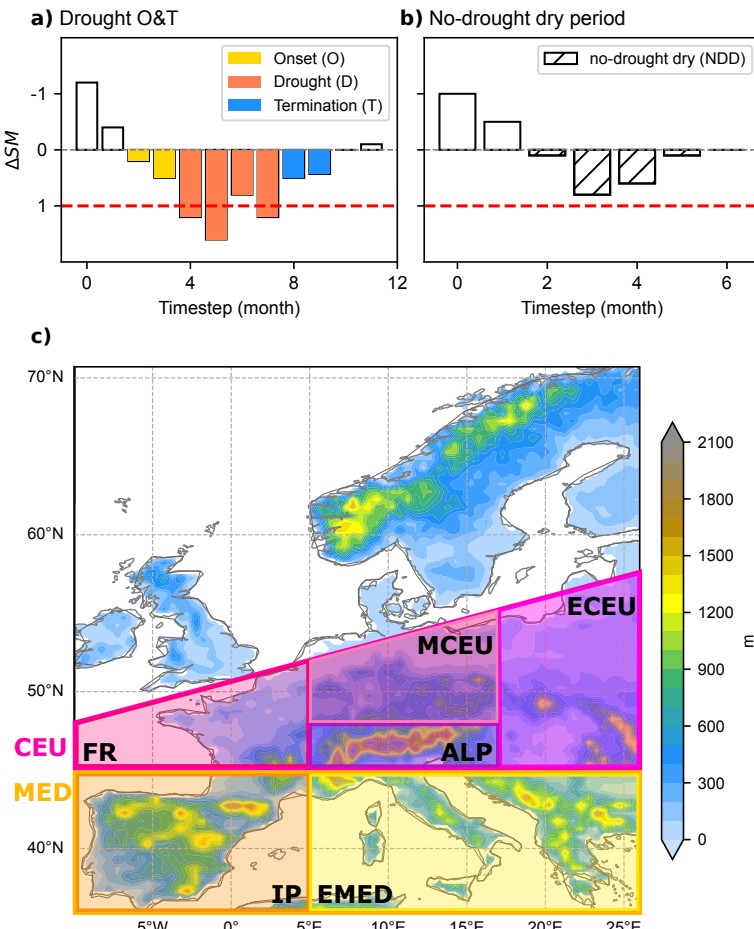

**Figure 1.** Illustration of drought phases based on the soil moisture anomaly ($\Delta SM$): a) a complete drought life cycle O, D, & T, b) a no-drought dry period. Shaded in colors are the onset (O; yellow), drought (D; orange), and termination (T; blue). The red-dashed lines indicate the $-1\sigma$ drought threshold. c) Topography in Europe and regional divisions based on Christensen and Christensen (2007) and Iturbide et al. (2020).

To examine the potential causes of discrepancies between the soil moisture datasets in representing the variability of $\Delta SM$

and drought phases, we compare evapotranspiration and precipitation from these datasets. For this, the monthly means of evapotranspiration and precipitation during the common period 2000–2020 are calculated over the study region, and also their anomalies are obtained by deseasonalizing the time series.

**3.4 Relationship between the duration of drought phases, drought intensities, precipitation, evapotranspiration**

To assess the relationship between the duration of onsets and droughts and between the duration of terminations and droughts,

first, the duration of individual O&T events from the grid points over the study domain (CEU and MED together) are collected.



Then, the individual durations of onsets or terminations, $d_{onset}$ or $d_{term}$, are grouped based on the same duration. The drought durations $d_{drought}$ within each onset group with $d_{onset} = i$ (or each termination group with $d_{term} = i$) are then spatially weighted and averaged. This results in a mean drought duration $\bar{d}_{drought}$ that belongs to $d_{onset} = i$ or $d_{term} = i$. For example, for the onsets, the mathematical expression for the spatially-weighted averaged mean drought duration $\bar{d}_{drought}$ for $d_{onset} = i$ is:

$$\bar{d}_{drought}(i) = \frac{\sum_{n=1}^{N} d_{drought}(n,i) * cos(lat(n))}{\sum_{n=1}^{N} cos(lat(n))} \quad n = 1, ..., N; \quad i = 1, ...M \tag{1}$$

where $d_{drought}$ is the durations of the individual droughts with the onset duration $i$, with $i$ ranging from 1 to the maximum duration $M$ of onsets, $N$ is the total number of grid points, and $lat(n)$ is the corresponding latitude for the grid point at location $n$. Eq. 1 is repeated for all $i$ until $M$, resulting in a series of mean drought durations with $M$ values. The same is repeated for drought durations for terminations. Here, instead of taking the common period among the datasets, the entire period of each dataset is used. Finally, $\bar{d}_{drought}$ and the O&T durations are compared.

Next, we assess the relationship between the O&T duration and the intensities of droughts to address whether intense droughts relate to the duration of onsets or termination. Drought intensities are estimated using the cumulative $\Delta SM$ that combines the mean decreases in $\Delta SM$ during droughts and the drought duration. The same Eq. 1 is repeated for all $d_{onset} = i$ and $d_{term} = i$, but replacing $d_{drought}$ by the cumulative $\Delta SM$ of drought events.

Lastly, we also examine how aspects of surface water balance, more specifically, precipitation (P), and precipitation minus evapotranspiration (P-E) are related to the O&T duration. Of particular interest is to identify whether the required net input water to initiate or terminate droughts depends on the duration of the transition periods, i.e., O&T. Alternative P and E datasets are used for those soil moisture products whose input P and resulting E are not openly available. For SoMo, P and E are taken from ERA5 and ERA5-Land, respectively. For GLEAM, P is obtained on the observation-based E-OBS, as the input forcing of GLEAM is a merged P from observation and satellite-based products.

Using these P and E, cumulative anomalies of P, $\Delta P$, and P-E, $\Delta(P-E)$ associated with $d_{onset}$ and $d_{term}$ are estimated. $\Delta P$ and $\Delta(P-E)$ are obtained by deseasonalizing P and P-E with respect to 2000–2014. The spatial averages of cumulative $\Delta P$ and $\Delta(P-E)$ corresponding to $\bar{d}_{onset} = i$ or $\bar{d}_{term} = i$ are computed repeating Eq.1, replacing $d_{drought}$ by cumulative $\Delta P$ and $\Delta(P-E)$. The relationship between the mean duration of onsets or terminations, cumulative $\Delta P$, and $\Delta(P-E)$ are assessed.

### 3.5 Circulation patterns and analysis of regional variables during drought phases

Atmospheric circulation patterns during O&T are depicted using anomalies of geopotential height at 500 hPa ($\Delta GP$) from ERA5. $\Delta GP$ is obtained by deseasonalizing the monthly geopotential height with the 2000–2014 multi-year annual cycle. Then, three-month running means of the anomalies are calculated in the same manner as was done to obtain $\Delta SM$.

We analyze atmospheric circulation anomalies during O&T's, considering O&T periods that the three soil moisture datasets, ERA5-Land, GLEAM, and CLM-TRENDY agree on during 1980–2020. When examining the atmospheric circulation, we focus on the atmospheric drivers of large-scale dryness, which affects the mean $\Delta SM$ of each subregion, as opposed to the



circulation patterns accompanying dryness at every grid point. Therefore, for this aspect of the analysis, O, T, and NDD are determined using the spatially averaged time series of $\Delta SM$ for each of the six subregions. Mean circulation during NDDs

is examined to understand which atmospheric conditions help onsets to progress into droughts and not be dissipated before reaching droughts as dryness during NDDs.

Lastly, the mean NAO index associated with each drought phase and NDD is determined to assess the influence of this dominant circulation pattern in the Euro-Atlantic region on drought phases. Also, the Pearson correlation coefficient between the NAO index and $\Delta SM$ is estimated over the entire period and each season at each grid point.

**4 Results and discussions**

**4.1 Temporal characteristics of drought onsets and terminations**

The duration of onsets and terminations (O&T duration) of droughts are estimated during 2000–2019 for SoMo and 2000–2020 for the other four soil moisture datasets. The mean duration is presented in Fig. 2. The duration of ERA5-Land, GLEAM, and CLM-TRENDY during 1980–2020 is included in the supplement (Fig. S1).

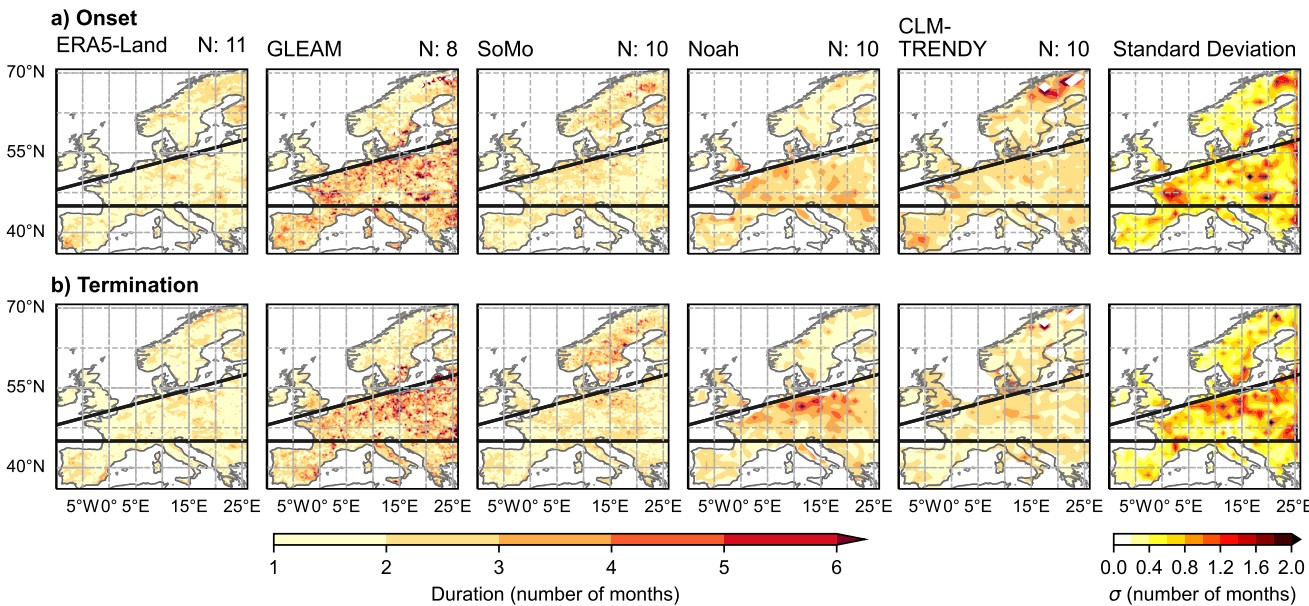

**Figure 2.** The mean duration of onsets and terminations of droughts over Europe during 2000–2019 for SoMo, and during 2000–2020 for the other four datasets. N indicates the median of the number of drought events over the study region. The standard deviations of the mean duration across the five soil moisture datasets are shown in the right-most column. Black thick lines separate central Europe (CEU) and Mediterranean Europe (MED) following the IPCC climate reference regions (Iturbide et al., 2020).



Fig. 2 indicates that within the same dataset, the mean O&T duration slightly varies across the regions in Europe. When the comparison is performed between the two spatial divisions, CEU and MED, most of the datasets tend to exhibit a slightly longer mean O&T duration in CEU compared to MED. The standard deviations across the datasets (shown in the rightmost column of Fig. 2) indicate that the discrepancies in the duration are more pronounced in CEU than in MED. In general, the difference in the mean O&T duration seems to be larger between the datasets than between the regions. It is noticeable that

ERA5-Land shows shorter and GLEAM shows longer O&T duration than others. Moreover, the datasets that present longer onsets also exhibit longer terminations, implying that the duration of onsets and terminations is related across the datasets.

To illustrate better the typical O&T duration in CEU and MED, the durations of all individual events are collected from the grid points in CEU and MED from Fig. 2 and presented in the box plots in Fig. 3a and b. For ERA5-Land, GLEAM, and CLM-TRENDY, the values for the entire period 1980–2020 are also included. The averaged O&T duration over the entire CEU

and MED are shown in Table 2.

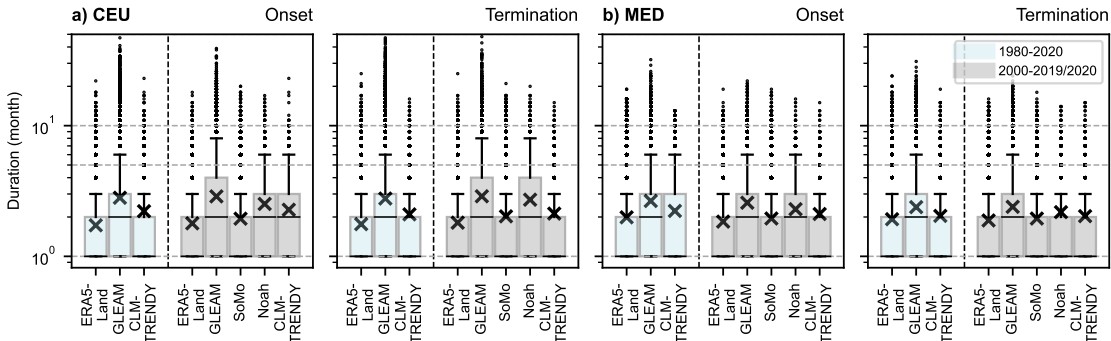

**Figure 3.** Duration of O&T of all individual events collected from the grid points over a) CEU and b) MED in Fig. 2 for each dataset. Black horizontal lines on the boxes indicate the medians and the crosses indicate the means of the duration. Note that the y-axis is on a logarithmic scale. Dashed grey horizontal lines indicate the duration of 1, 5, and 10 months.

Comparing the long (1980–2020) and short (2000–2020) time series, the mean O&T duration between both periods are similar in each ERA5-Land, GLEAM, and CLM-TRENDY (Fig. 3 and Table 2). Considering the period 2000-2019/2020, ERA5-Land shows the shortest mean O&T duration in both regions, with the mean values over CEU and MED ranging from 1.79 to 1.88 months (Table 2), followed by SoMo (1.94 to 2.02 months) and CLM-TRENDY (2.03 to 2.27 months). GLEAM

exhibits the longest O&T duration, from 2.39 to 2.88 months. In general, the 75th percentile values of the datasets range from 2 to 4 months.

The duration between O&T within the same dataset is generally related. The datasets with short onset also exhibit short termination. Statistically, GLEAM (both periods, 1980–2020 and 2000–2020), ERA5-Land (2000–2020), and CLM-TRENDY (both periods) in CEU, SoMo, Noah, and CLM-TRENDY (2000–2020) in MED indicate that the O&T duration is the same

(Table 2). Although the O&T duration for the rest is statistically different based on the t-tests, the magnitudes of O&T duration within the same dataset are relatively similar compared to the difference in the O&T duration between the datasets. The





**Table 2.** Mean duration of O&T (in number of months) from Fig. 3a and b. When the duration of onsets and terminations within the same dataset and region are statistically different from each other based on the t-tests at a 95% confidence level, the values are denoted with *.

| | | 1980-2020 | | | 2000-2019 | 2000-2020 | | | |
| | | ERA5-Land | GLEAM | CLM-TRENDY | SoMo | ERA5-Land | GLEAM | Noah | CLM-TRENDY |
|---|---|---|---|---|---|---|---|---|---|
| **CEU** | Onset | 1.73* | 2.81 | 2.22 | 1.94* | 1.79 | 2.88 | 2.52* | 2.27 |
| | Termination | 1.76* | 2.77 | 2.09 | 2.02* | 1.88 | 2.88 | 2.71* | 2.12 |
| **MED** | Onset | 1.99* | 2.65* | 2.23* | 1.95 | 1.84* | 2.57* | 2.30 | 2.11 |
| | Termination | 1.92* | 2.38* | 2.04* | 1.95 | 1.88* | 2.39* | 2.19 | 2.03 |

result implies no noticeable differences in the time required to progress into droughts and recover from droughts. Still, larger discrepancies exist in the duration of either onsets or terminations between the datasets. As presented in Figs. 2 and 3, all datasets exhibit different values for the duration, with GLEAM presenting the longest duration for both O&T.

Regarding the preferred seasons for O&T, the ratios of the number of onsets for each season to the total number of onsets during the 1980–2020 period, as defined in Section 3.3, are presented for ERA5-Land, GLEAM, and CLM-TRENDY in Fig. 4, and for terminations in Fig. 5. Similar to the duration, Fig. 4 and Fig. 5 show that there are regional differences within Europe regarding the occurrence timing of O&T. The majority of areas in CEU indicate JJA as the most frequent season for onsets, although not uniformly over the region and with some inconsistency in the ratios among the datasets (Fig. 4). GLEAM exhibits

higher ratios distributed across extensive areas in CEU than ERA5-Land and CLM-TRENDY during JJA. Southern CEU, mainly ALP, shows higher occurrence ratios during SON in ERA5-Land and GLEAM, and during DJF in CLM-TRENDY. Over IP, onsets tend to occur more frequently during SON and DJF for all three datasets, whereas in the EMED, over Italy and the Balkans, they are most likely to occur in SON. Over Sardinia, Corsica, and northern Italy, onsets also occur during DJF in a more pronounced manner in CLM-TRENDY, but this is not evident in GLEAM.

For terminations (Fig. 5), more areas in CEU experience terminations during DJF and MAM. ECEU also exhibits a high frequency of terminations during SON. However, the occurrence ratios during terminations show more differences across the datasets than during onsets. For instance, ERA5-Land also exhibits a higher occurrence ratio for termination (> 0.3) in MCEU during MAM. However, this is not as apparent over the same region in GLEAM and CLM-TRENDY. This difference between the datasets also happens in MED. Over MED, higher occurrence ratios for terminations are observable during MAM in ERA5-

Land and CLM-TRENDY, but this is not visible in GLEAM. In general, over IP, drought terminations occur frequently during DJF, MAM, and JJA. The majority of areas in EMED experience terminations during DJF and MAM. In general, most regions in MED agree on a relatively low occurrence of terminations during SON.

  Fig. 6 summarizes the frequencies of onsets and terminations seen in Figs. 4 and 5 for each subregion by showing the median of the distribution of ratios across each domain. For onsets, Fig. 6a implies that although more subregions are considered, the

seasonality of onsets over the study region can be largely divided into two spatial domains: CEU and MED. Over CEU (FR, MCEU, ALP, and ECEU), JJA is the most likely season for onsets in ERA5-Land and GLEAM, and SON in CLM-TRENDY.



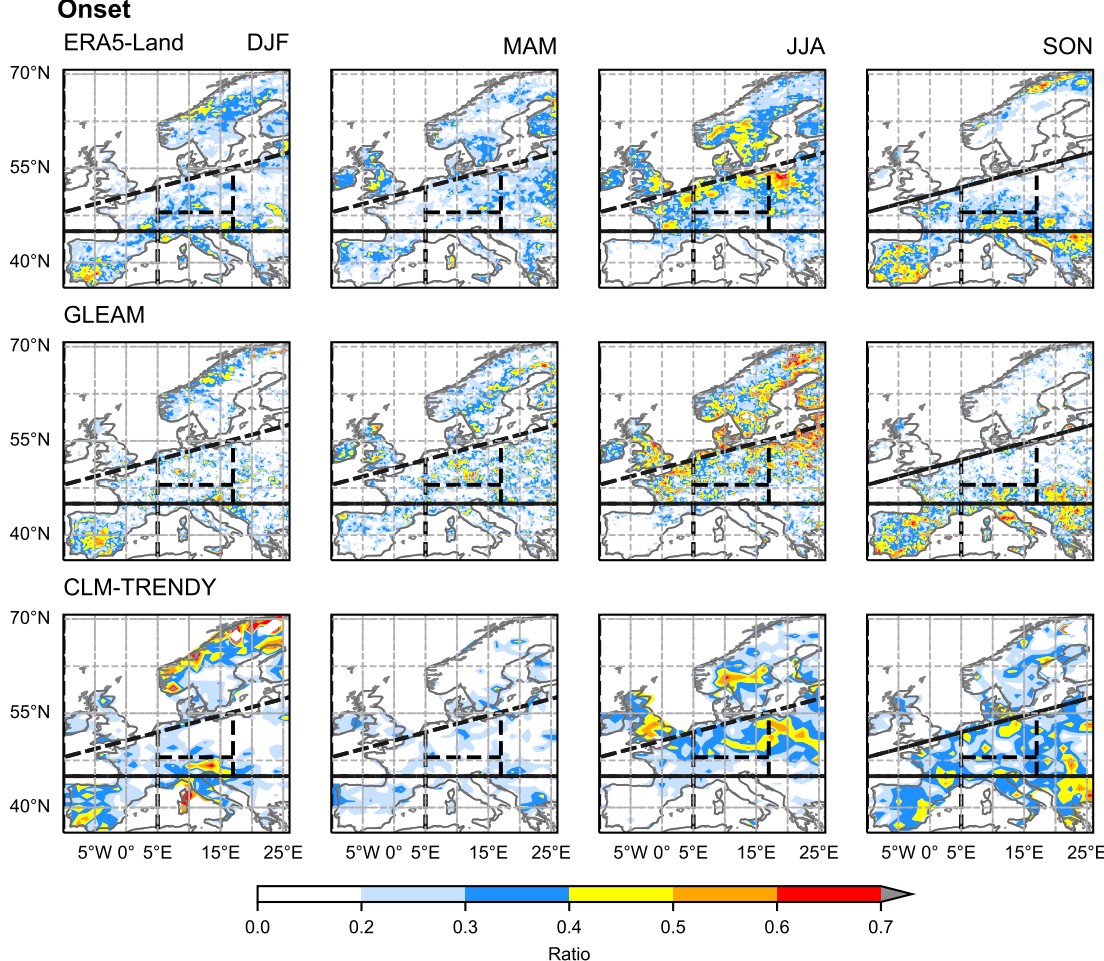

**Figure 4.** Ratios between the number of onsets (last month of onsets) for each season and the total number of onsets during 1980–2020 defined in Section 3.3. Black lines separate the continent into the subregions: Iberian Peninsula (IP), eastern Southern Europe (EMED), Alps (ALP), France (FR), mid-Europe (MCEU), and east central Europe (ECEU).

The most unlikely seasons range between DJF and MAM, varying across the regions and datasets. For MED (IP and EMED), all datasets unanimously indicate that the most likely season onsets is SON, followed by DJF. The least likely season for onsets is JJA.

The occurrence ratios of terminations do not present a robust spatial division between the two regions as onsets, and there are some discrepancies among the datasets on the seasonality of the occurrence (Fig. 6b). In CEU, over MCEU, ALP, and ECEU, GLEAM and CLM-TRENDY indicate DJF as the most likely season for terminations. For FR, GLEAM still denotes DJF, but CLM-TRENDY points to MAM as the most likely season. Nevertheless, DJF is the second most likely season in CLM-TRENDY, and the difference in the occurrence ratio between DJF and MAM is small (< 0.01). In ERA5-Land, MAM is



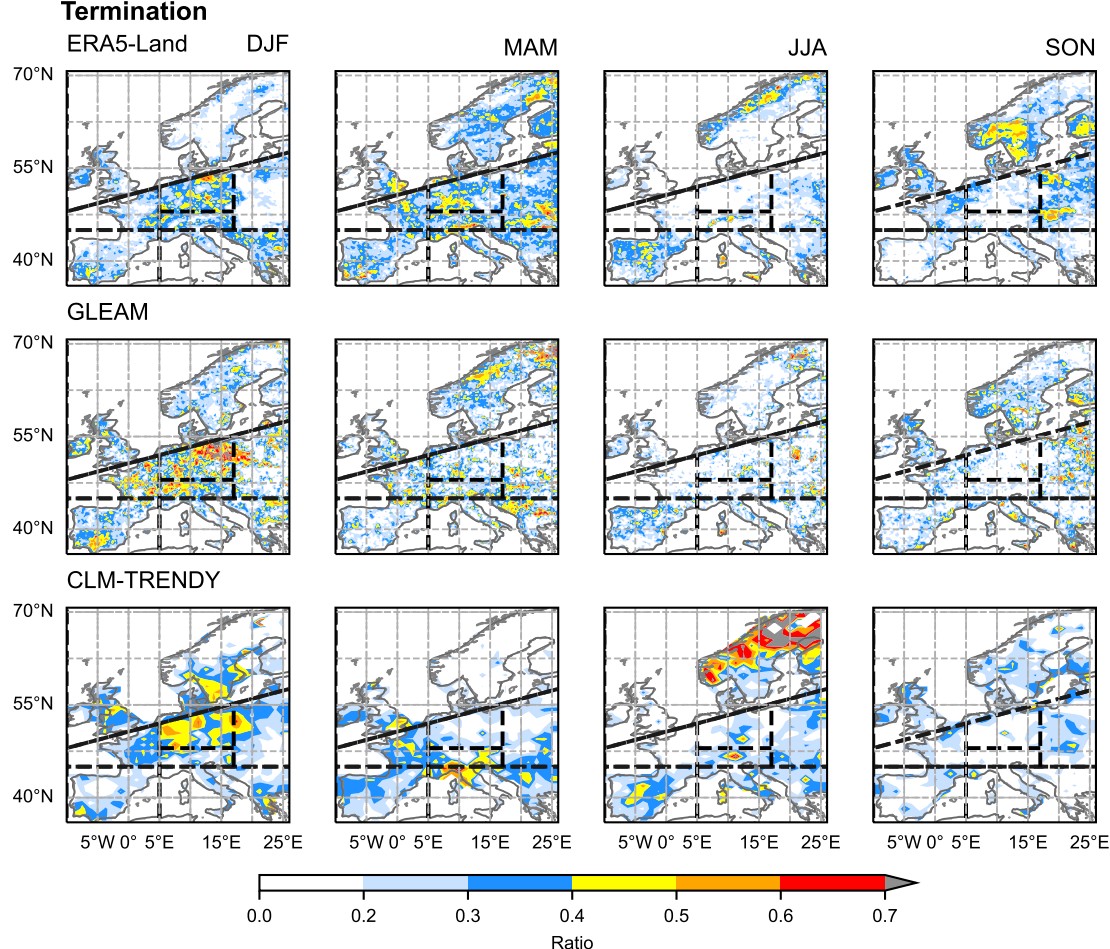

**Figure 5.** Same as 5, but for terminations (first month of terminations).

the most likely season for terminations in all regions of CEU. In FR, MCEU, and ALP, DJF is the second most likely season. In MED, over EMED, all datasets point out MAM as the most likely season for terminations. In IP, the discrepancy between the datasets for the preferred seasons for terminations is more pronounced. ERA5-Land exhibits JJA (followed closely by MAM), GLEAM presents DJF (followed by JJA), and CLM-TRENDY shows both MAM and JJA as preferred seasons for terminations.

In general, the preferred seasons for onsets coincide with each region's wet seasons. For CEU (FR, MCEU, ALP, and ECEU), these periods are JJA, and for MED (IP and EMED), these seasons correspond to SON and DJF, as indicated by their annual cycles, shown in Fig. 7. On the other hand, terminations tend to occur more frequently during the non-wet seasons, which are DJF and MAM in CEU, and MAM and JJA in MED. This result highlights the crucial role of precipitation availability and



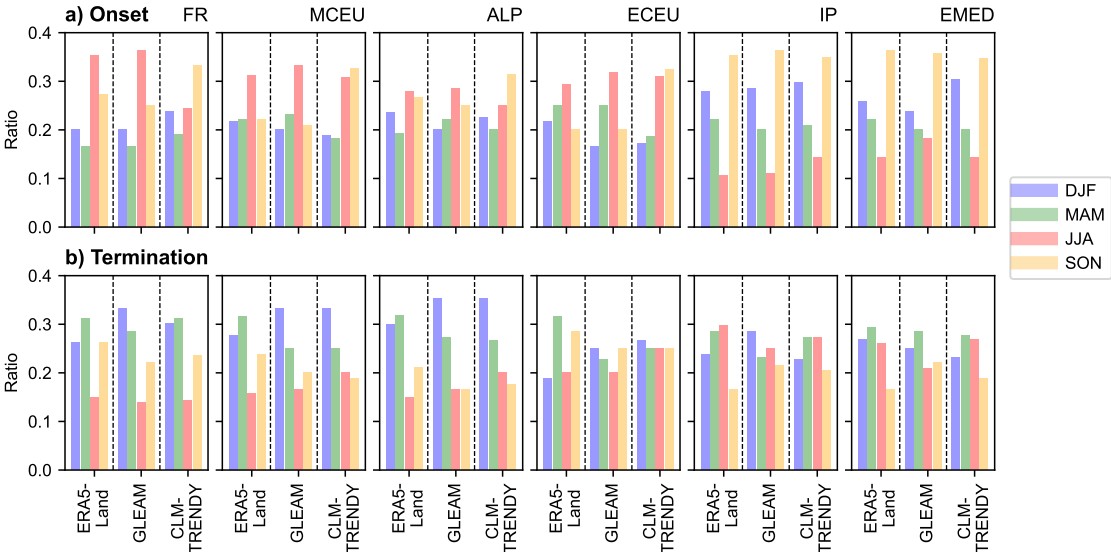

**Figure 6.** a) The medians of occurrence ratio of onsets for each season over all grid points in each domain from Fig. 4. (b) is the same as (a) but for terminations.

related circulations during the wet seasons in initiating dry conditions leading to drought onsets, while drought terminations
are not particularly attached to the wet seasons.

Nevertheless, it is important to remark that while there are some preferred seasons for O&T showing higher occurrence ratios, O&T can still occur in other seasons, as seen in Fig. 6. The occurrence ratio of the most likely seasons does not overly exceed the ratio during other seasons. For example, in IP for ERA5-Land, the ratio for SON is 0.35, while for DJF is 0.28. The occurrence ratios for O&T are approximately 0.2 for the majority of the seasons and datasets, except during JJA in IP.

**4.2    Differences in soil moisture, precipitation, and evapotranspiration among the datasets**

To examine the potential origin of the differences in the duration and the preferred seasons for O&T among the datasets, we perform a comparison between the time series of $\Delta SM$ and the means of precipitation (P) and evapotranspiration (E) across the LSMs. As seen before, there are, in general, two spatial separations in the duration and seasonality of O&T (Figs. 2 and 6), CEU and MED. Thus, the study region is divided into these two to generate the time series. The time series of $\Delta SM$ for each
dataset are presented in Fig. 8.

The difference in the $\Delta SM$ over time among the datasets appears to be more pronounced in CEU than MED, which is consistent with the standard deviations presented in Fig. 2. A noticeable difference is observed between GLEAM and the other two datasets before 1991. During this period, GLEAM constantly exhibits positive mean $\Delta SM$ over CEU. Regarding the dry condition since 2016, GLEAM and CLM-TRENDY present constant negative $\Delta SM$, while ERA5-Land shows some months
with positive $\Delta SM$. The difference among the datasets is smaller in MED than in CEU, with smaller $\sigma$.







**Figure 7.** Annual cycles of precipitation during the reference period 2000–2014 from ERA5, Noah forcing, E-OBS, and CLM-TRENDY forcing for a) CEU and b) MED. (c) Annual mean precipitation during the same period over Europe.

E is a variable involved in the terrestrial water balance and is internally estimated by the LSMs, although it is influenced by aspects of the observational forcing as well, such as near-surface specific humidity and wind speed. Hence, the difference in E between the datasets is most likely indicative of a difference in the model's internal physics. Evapotranspiration during 2000–2020 is shown for each of the datasets in Fig. 9.





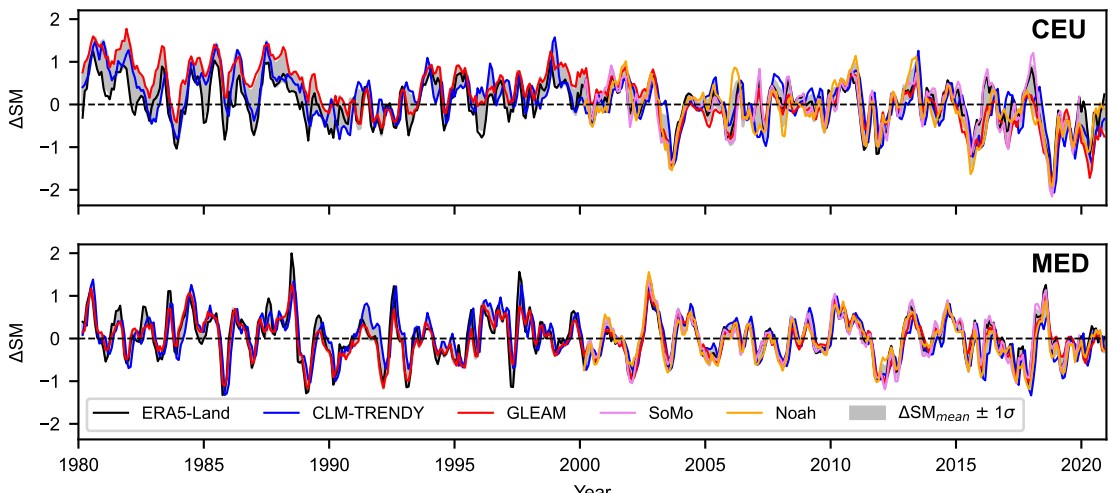

**Figure 8.** Spatially-weighted time series of $\Delta SM$ over CEU and MED. Grey shading denotes $\Delta SM_{mean} \pm 1\sigma$. $\Delta SM_{mean}$ is the average and $\sigma$ is the standard deviation of $\Delta SM$ across the five datasets.

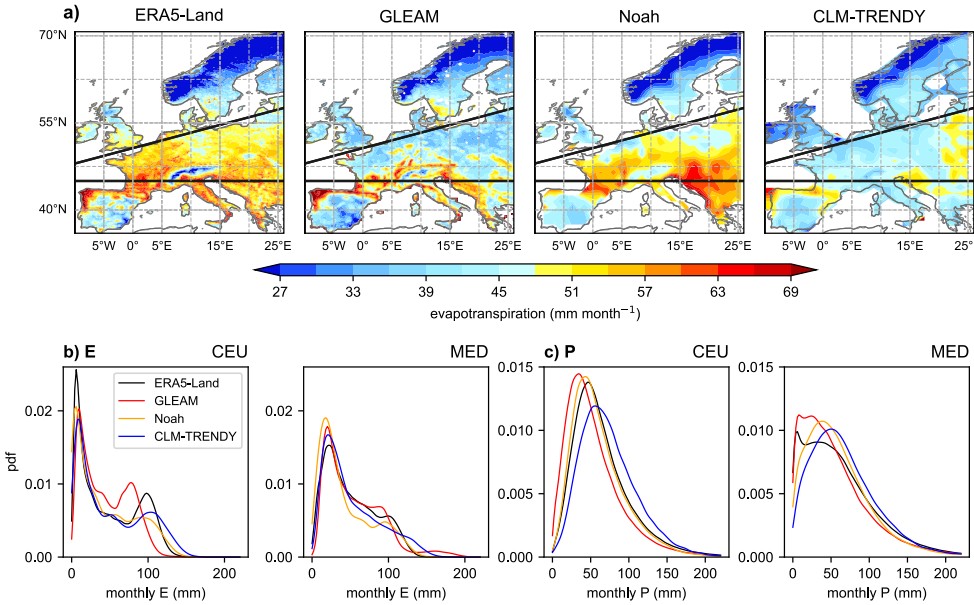

**Figure 9.** a) Monthly mean evapotranspiration (E) over 2000–2020 in Europe, b) probability density distributions of monthly mean evapotranspiration, and c) mean precipitation (P) over CEU and MED for the same period. SoMo is excluded as it does not contain the evapotranspiration variable.



Similar to the duration and preferred seasons, E varies across the regions within Europe and it also varies across datasets. In terms of the datasets, ERA5-Land and Noah exhibit higher mean E than others. GLEAM has higher values over southern CEU and northern IP. CLM-TRENDY tends to show reduced mean E compared to other datasets. GLEAM presents a longer duration of O&T than other datasets (Fig. 2 and Table 2), which can be related to reduced mean evapotranspiration due to relatively longer dry periods. In the case of CLM-TRENDY, reduced E can be caused by relatively reduced mean P compared to other datasets (Fig. 9c), primarily over CEU.

For P (Fig. 7c), which is the input forcing for LSMs, some regional differences exist among the datasets, with the largest differences over the alpine region. Nevertheless, the four datasets, in general, present consistent spatial patterns of total mean precipitation. The differences in the means are not as remarkable as evaporation, as shown in Fig. 9a. Over MED, the probability density distributions of P (Fig. 9c) indicate that some differences exist in the density distributions among the datasets, including the maximum peak of precipitation. Over CEU, CLM-TRENDY shows the P distribution shifted toward a higher mean.

       Overall, the comparison of $\Delta SM$, E, and P among the datasets suggests that varying temporal variability of $\Delta SM$, mean E, and P contribute to the observed discrepancies in the duration and seasons among the datasets. In summary, both the models' internal physics and the ingested atmospheric forcing, i.e., P, are the contributing factors (Fang et al., 2016). Note that other variables involved in soil water balance could have been considered for the analysis, but only evapotranspiration was chosen

since it is the variable available in all four soil moisture products. Obviously, there are other variables involved in soil water balance not shown here, such as surface runoff.

       In spite of these differences, some similarity in the temporal variability in $\Delta SM$ is observed (Fig. 8), and the time series are significantly correlated to each other (not shown). This indicates a certain range of similarity in temporal variability exists between the datasets, inducing similar preferred seasons of occurrence.

### 4.3  Relationship between the duration of onsets and terminations, drought intensities, precipitation, and evapotranspiration

Next, we want to address whether the drought duration is related to the O&T duration. Do long (short) droughts also have long (short) onsets and/or terminations? The overall relationships between the duration of drought phases are presented in Fig. 10. For this, all the drought durations that correspond to a certain value of onset and termination duration are collected from all

grid points over the study domain and then averaged, taking into account the spatial weights following eq. 1.

       Fig. 10a and b indicate that the duration of O&T and droughts do not present a robust relationship. Longer (shorter) droughts do not necessarily have longer (shorter) O&T. Only CLM-TRENDY indicates that the drought duration is statistically significant and positively related to the onset duration, with a month of onset leading to an increase of 0.28 months in the drought duration. GLEAM exhibits a significant relationship in the duration between droughts and terminations, with an increase of

0.30 drought months every month of termination. In general, there are no unanimous directions, which means the same signs, of relationship among datasets. This result suggests that the onset duration is not indicative of the longevity of droughts, and the drought duration does not affect the termination duration.



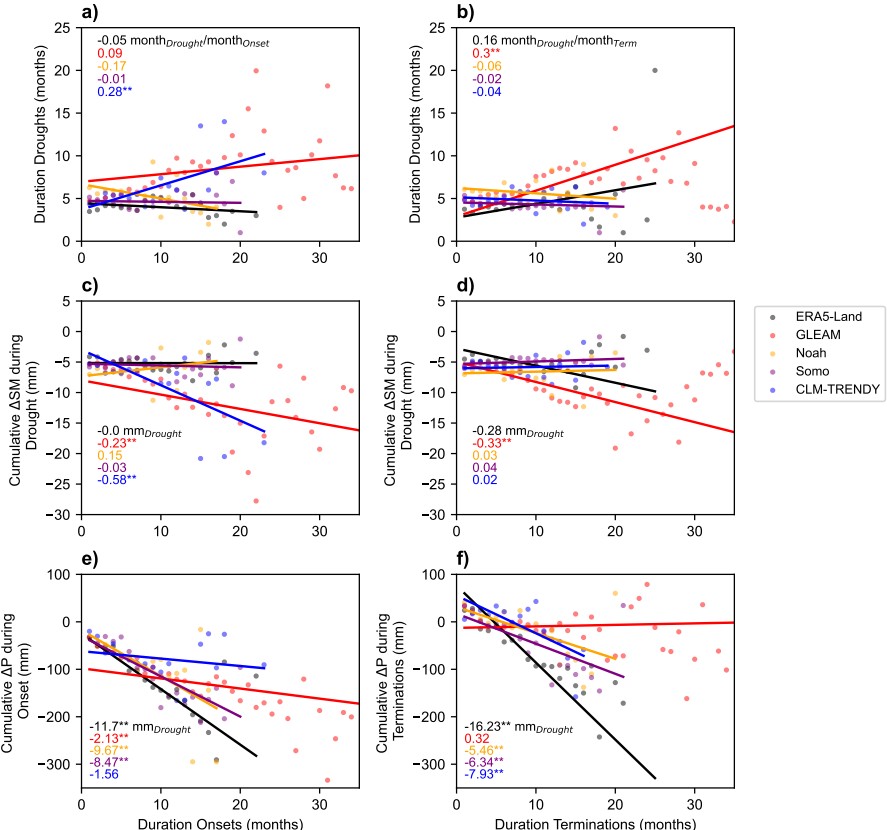

**Figure 10.** Relationships between (a) the mean duration of onsets and droughts, b) between the mean duration of terminations and droughts. c) and d) Same as (a) and (b) but between mean duration of O&T and drought intensities expressed in cumulative soil moisture anomalies. e) The mean duration of onsets and cumulative $\Delta P$ during the onsets, and f) the mean duration of terminations and cumulative $\Delta P$ during the terminations. The mean duration of droughts and cumulative $\Delta P$ are calculated following Section 3.4. Thick solid lines indicate the coefficients from the linear regression models between the two variables. The numbers indicate the regression coefficients denoting the degree of the relationship. When the estimated coefficients are statistically different from zero at a 95% confidence level based on the Wald test, the values are denoted with **.

The same finding is observed in terms of intensities of droughts (Fig. 10c and d). Intensities of droughts, represented as cumulative $\Delta SM$ during droughts (more negative cumulative $\Delta SM$, more intense the drought), and the duration of either onsets or terminations do not consistently show statistically significant relationships in all datasets. Only GLEAM and CLM-TRENDY exhibit statistically significant relationships, indicating decreases of 0.23 mm and 0.58 mm in soil moisture every month of onset, respectively. For terminations, GLEAM is the only dataset showing that intense droughts have longer termination (a decrease in 0.33 mm of soil moisture linked with a month of termination).



How $\Delta P$ progressing during onsets and terminations are linked to the O&T duration is also examined (Fig. 10e and f). This
analysis is to evaluate whether the magnitude of changes in $\Delta P$ can signal a drought initiation or a potential drought termination
and, consequently, an early warning based on this relationship. For onsets, all datasets indicate negative relationships between
the onset duration and the cumulative $\Delta P$; among them, four datasets (ERA5-Land, GLEAM, Noah, and CLM-TRENDY)
denote statistically significant relationships. The duration of terminations also presents negative relationships to the cumulative
$\Delta P$, with four datasets (ERA5-Land, Noah, SoMo, and CLM-TRENDY) exhibiting statistically significant values. Longer
terminations mean more periods in dry conditions where cumulative negative $\Delta P$ increases. Hence, a longer termination does
not necessarily mean more periods of precipitation that lead to the end of droughts. A similar result is observed when instead
of cumulative $\Delta P$, cumulative $\Delta(P\text{-}E)$ is analyzed (Fig. S3 in the supplement), meaning that the magnitude of $\Delta P$ (or $\Delta(P\text{-}E)$)
gives a prior indication of the duration of O&T.

Overall, the result indicates that the duration and intensities of droughts do not give a specific indication of the duration
of onsets and terminations. On the other hand, in terms of the water balance conditions, $\Delta P$ and (or $\Delta(P\text{-}E)$) become more
negative with longer onsets and terminations. This means that $\Delta P$ can potentially offer an early signal regarding the potential
duration of drought onsets and terminations, given its magnitudes.

## 4.4  Atmospheric circulations associated with onsets and terminations

To understand the atmospheric circulation involved in each drought phase and which circulation conditions help onsets to
progress into droughts, the mean circulation patterns during onsets, terminations, and no drought dry periods (NDDs) are
examined and are presented in Fig. 11.

In all onsets, the mean circulation patterns (Fig. 11a) are characterized by an anticyclonic circulation and negative $\Delta P$
over the regions where droughts take place. Although the exact location of the system slightly differs, large-scale anticyclonic
pressure systems in Europe during onsets resemble the circulation patterns that were observed during the recent European
droughts (Ionita et al., 2017; García-Herrera et al., 2019) and the pattern associated with a strong reduction in precipitation
over the region affected by droughts (Gessner et al., 2022). This system can further decrease precipitation by inhibiting the
incoming moisture from the Mediterranean Sea and the Atlantic Ocean since it is associated with weakened westerlies.

During NDDs (Fig. 11b), similar anticyclonic patterns occur over extensive areas in Europe, also with negative $\Delta P$. How-
ever, we found that the magnitude of $\Delta GP$ over the affected region is lower during NDDs than those of onsets. In addition,
the occurrence of positive $\Delta GP$ is less frequent during NDDs, indicating that the persistence of the anticyclonic system is
required to make dry conditions progress into droughts.

During terminations (Fig. 11c), ECEU and EMED show a cyclonic pattern over northern Europe. For other regions, the mean
circulation patterns do not show well-defined structures over the region where termination takes place, unlike onsets, implying
that opposite cyclonic conditions over the affected regions are not strictly required for terminations. Instead, a high-pressure
system located over the North Atlantic Ocean is observed during terminations in all regions except in ECEU. Anticyclonic
circulation associated with this pressure system in the North Atlantic can bring moisture from the higher latitudes to the
continent. Terminations occurring in MED (IP and EMED) show positive $\Delta GP$ in the southern regions (about below $45°$) and



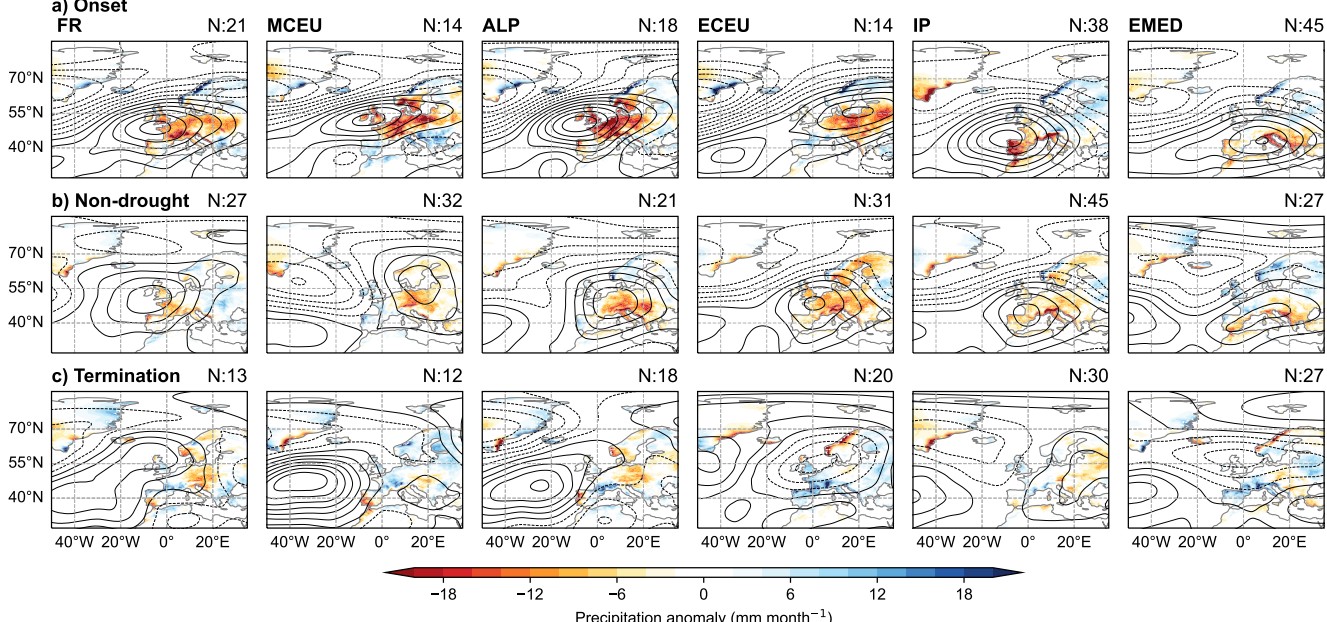

**Figure 11.** a) Mean geopotential height anomalies at 500 hPa ($\Delta GP$) during drought onsets (in contour lines every 6 gpm, where the dotted lines denote negative anomalies) and $\Delta P$ (color-shaded) associated with considered onsets for each of the subregions, b) and c) Same as (a) but averaged over that season in no-drought dry periods and terminations, respectively, following Section 3.5. Drought phases for a-c are defined using the spatially weighted averaged time series of $\Delta SM$ for each subregion. N indicates the number of timesteps considered for the composites.

negative $\Delta GP$ in the north. In general, a weakened $\Delta GP$ is observed over each region of interest, and positive $\Delta P$ is evident during the termination phase.

Similar results of anticyclonic circulation during onsets and NDDs and weakened $\Delta GP$ during terminations are also found for seasonal analysis. A slight difference is observed in the location of the anticyclonic patterns during onsets and NDDs depending on the season (Figs. S4 to S6).

To assess the overall relationship between drought phases and NAO, the correlation between the NAO index and monthly or seasonally averaged soil moisture anomalies is presented in Fig. 12a and b. Fig. 12a indicates that when considering all months,
NAO is negatively correlated with soil moisture over CEU and MED, suggesting a positive NAO associated with a decrease in soil moisture over CEU and MED. Although it is not within the regions of our interest, soil moisture in the Scandinavian region is positively correlated to NAO. These relationships between soil moisture and NAO agree with Almendra-Martín et al. (2022), who showed a dominant influence of the NAO on soil moisture variability in Europe. However, the correlation coefficients vary with seasons. During JJA, the positive correlation coefficients dominate over a large portion of MED, and
during SON in southeastern IP. The correlation coefficients are positive during DJF over MCEU and ECEU. The result suggests a varying influence of NAO on soil moisture droughts depending on the season. In general, over MED, the effect of NAO on



soil moisture droughts is strong during DJF and MAM, and also during SON over EMED, with negative correlations. The correlations become weaker during JJA. For CEU, the correlations are negative during MAM and JJA. EMED shows more negative correlations during SON. The correlations become positive during DJF in MCEU, indicating less influence of NAO on soil moisture droughts in the region, while the correlations are negative in FR. These seasons seem to be consistent with the preferred seasons for onsets and terminations of each region shown in Fig. 6. It is noticeable that the Alpine region exhibits a clear opposite correlation pattern to the rest of the regions, which emphasizes the complex topography of this area.

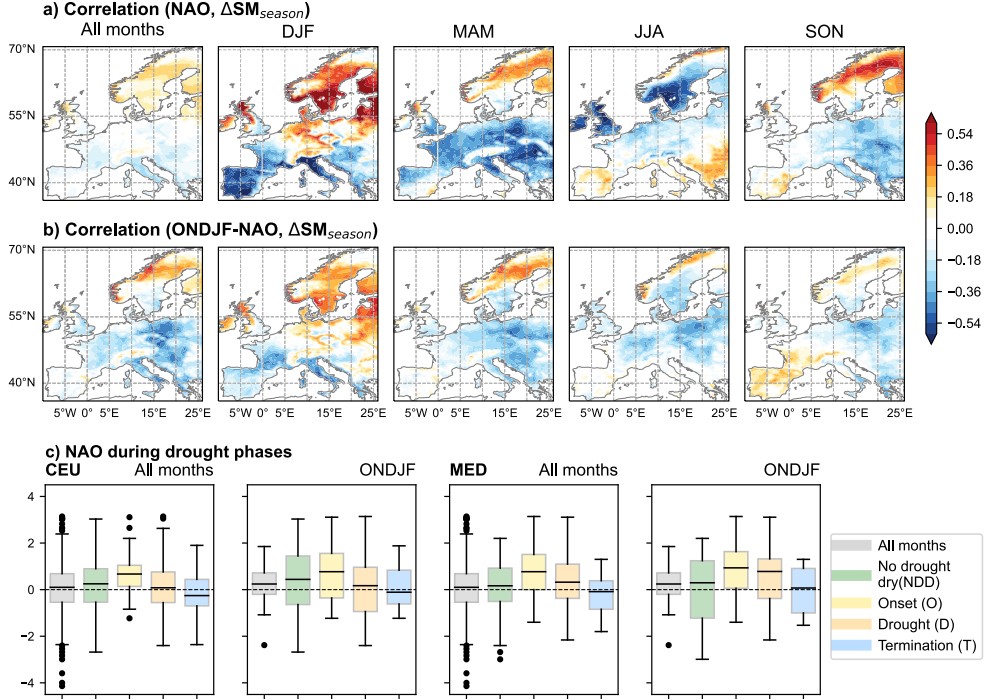

**Figure 12.** a) Pearson correlation coefficients between the NAO index and monthly or seasonally averaged soil moisture anomalies ($\Delta SM_{season}$) for all periods and each season, and b) between the cold-season (October-February) NAO indices and $\Delta SM_{season}$ for each season. c) NAO index values during each drought phase and NDD. The medians are marked in black lines. Drought phases are estimated using the spatially-weighted time series of $\Delta SM$ in Fig. 8.

Knowing that the strength of NAO increases during the cold seasons, the correlation coefficients are also calculated between the mean October-February NAO index and soil moisture anomalies, which is shown in Fig. 12b. In all seasons, the correlation patterns are similar to those in Fig. 12a. The difference is that during DJF, the positive correlations are extended toward ECEU and Balkan regions. During MAM and JJA, the magnitudes of coefficients are reduced over MED. During SON, positive coefficients are expanded to northern IP and FR.

To focus on the influence of NAO during each drought phase, the monthly and ONDJF NAO values are separated into drought phases and NDDs in CEU and MED and presented in the box plot in Fig. 12c. The mean NAO values for each phase





are shown in Table 3. For both the monthly and the cold-season NAO, positive NAO tends to occur more frequently during onsets than other phases. Persistent anticyclonic circulation observed during onsets in Fig. 11a can be associated with more frequent positive NAO, which leads to a weakening of westerlies as a moisture source from the Atlantic. In MED, positive NAO occurs more frequently during onsets and also during droughts.

In terms of the statistical difference in the mean values of NAO between the drought phases (Table 3), for MED, the mean NAO during onsets differs statistically from the mean NAO during the other phases. The result indicates that positive NAO occurs more frequently during onsets than other phases. The same result is obtained with the cold-season NAO during all drought phases, except for droughts. In the case of CEU, the difference between the mean NAO during onsets and other phases is statistically not pronounced when considering the cold-season NAO. However, considering all monthly NAO, the means are statistically different between onsets and all phases except for NDD.

Despite no statistical difference in some phases, it is clear that the mean NAOs are more positive during onsets than other phases in both regions. The differences in the mean values of NAO between onsets and NDDs are large (0.38 to 0.82), suggesting that persistent anticyclonic during onsets could be related to the frequent positive NAO. Terminations show more neutral or slight negative NAO, implying that a negative NAO condition is not strictly necessary to terminate droughts. Overall, the result shows that positive NAO occurs more frequently during the initial dry conditions preceding droughts, implying the role of NAO as an early warning of droughts.

**Table 3.** Mean NAO index values from Fig. 12c. When the mean NAO is statistically different from those during onsets based on the t-tests at 95% confidence level, the values are denoted with *.

|  |  | All | Onset | Drought | No drought dry | Termination |
|---|---|---|---|---|---|---|
| **CEU** | All months | 0.10* | 0.65 | 0.15* | 0.27 | -0.07* |
|  | ONDJF | 0.19 | 0.76 | 0.14 | 0.37 | 0.15 |
| **MED** | All months | 0.10* | 0.79 | 0.35* | 0.07* | -0.17* |
|  | ONDJF | 0.20* | 0.83 | 0.52 | 0.01* | -0.11* |

## 5 Discussions and conclusion

We have examined temporal characteristics, namely the typical duration and the seasons of occurrence, and atmospheric circulation associated with drought onsets and terminations (O&T) in central (CEU) and Mediterranean Europe (MED) from 1980 to 2020. Soil moisture droughts are quantified using the upper 10 cm soil moisture anomalies from five different soil moisture datasets: ERA5-Land, GLEAM, SoMo, Noah, and CLM-TRENDY. The drought life cycle is divided into three phases: onset, drought, and termination. Onsets and terminations are the transition periods to droughts and from droughts to normal conditions, respectively. This means that onsets are defined as the dry periods preceding the drought threshold, while terminations are the periods from the drought threshold to the neutral conditions.




We found that there are some differences in the duration of O&T across Europe and among the five soil moisture datasets used in this study. In general, CEU tends to exhibit a longer duration for both O&T compared to those over MED. In terms of the datasets, while ERA5-Land shows a shorter O&T duration in all phases, GLEAM exhibits a longer duration than other datasets. However, within the same datasets, the difference between O&T duration is relatively small, and for some datasets and periods, without any statistical difference. This suggests that the range of durations of onsets and terminations are similar.

Regarding the seasons of occurrence of O&T, the wet seasons are the most likely periods for onsets in CEU and MED. These are summer (JJA) for CEU and autumn (SON) and winter (DJF) for MED. This emphasizes the importance of precipitation availability and related atmospheric circulation during the wet seasons in initiating droughts. For terminations, there are no consistent seasons of occurrence among the datasets, but in general, the most likely periods for terminations occur during non-wet seasons. Nevertheless, although there are some preferred seasons for onsets, they can still take place in other seasons, and the frequency of occurrence during the most likely seasons does not overly exceed the frequency during other seasons.

The observed discrepancies in the duration and seasonality of O&T among the datasets seem to be partially explained by LSM's internal physics and input atmospheric forcings (Fang et al., 2016). This discrepancy needs to be considered when drought studies are based on a single soil moisture dataset. Still, the existence of some common seasons of occurrence of drought phases implies a certain range of similarity in the temporal variability of soil moisture across different LSMs.

By linking precipitation anomalies and the O&T duration, we found that the magnitudes of cumulative precipitation deficits are straightforwardly related to the duration of onsets or termination. Hence, the variable can serve as a direct indication of potential O&T duration.

In terms of atmospheric circulation anomalies, drought onsets are associated with anticyclonic atmospheric circulation patterns, a finding that agrees with Lhotka et al. (2020). For terminations, there are not unanimous patterns that different subregions in Europe share in common. In general, terminations are linked with circulation patterns with reduced magnitudes without a well-defined structure over the region where droughts occur. Terminations occurring over eastern Central Europe and the eastern Mediterranean, cyclonic circulation that occupies central and northern Europe is observed. Circulation patterns during onsets are stronger and more persistent than those during the normal light dry periods.

In addition, during onsets, the mean NAO is positive, indicating more frequent positive NAO than during other drought phases. The mean NAO during onsets is statistically higher than in other drought phases, including droughts, and the effects of NAO are more pronounced in MED than CEU. This indicates the important role of this large-scale circulation pattern during onsets in initiating a reduction of moisture supply and persistence of anticyclonic patterns during the most likely seasons of drought onsets. The finding remarks on the potential usage of the NAO index during onsets as an early warning for droughts. The mean NAO during terminations is in a neutral state.

Our study is one of the few that have investigated the onset and termination of droughts in Europe on a pan-continental scale over central and Mediterranean regions, providing clear, distinct temporal and circulation characteristics involved in different drought phases. A downside of this study would be that the temporal extent of the datasets, 20 to 40 years, may not be long enough to capture all possible variability of drought onsets and terminations.



It is also important to remark that regional differences in the temporal characteristics of onset and termination are observed within CEU and MED. Europe is characterized by complex topography and is located in the transition zone from a semi-arid
climate in the south to a relatively temperate wet climate in the north. Therefore, each climate may present different drought characteristics, which are related also to the seasonality of precipitation. Our analysis separates regional domains considering these climate aspects by largely following Christensen and Christensen (2007). However, it may not take into account very small-scale regional differences within these domains.

More understanding of onsets and terminations is certainly necessary to improve early predictions and preparedness for
potentially devastating droughts. How the life cycle of droughts is influenced by anthropogenic warming would be the follow-up research question that needs to be addressed to understand how these extreme hydrological events might change in the future.

*Code availability.* The python script with a function for drought phases estimation is available on the corresponding author's GitHub https://github.com/wmk21/drought_estimation.

*Data availability.* The datasets used in this study are available online in: ERA5 and ERA5-Land at https://cds.climate.copernicus.eu/, GLDAS LSMs at https://ldas.gsfc.nasa.gov/gldas, GLEAM at https://www.gleam.eu/, E-OBS at https://www.ecad.eu/download/ensembles/download.php, and SoMo.ml at https://www.bgc-jena.mpg.de/geodb/BGI/somo_ml_v1.php. The monthly NAO index is obtained from https://climatedataguide.ucar.edu/climate-data/hurrell-north-atlantic-oscillation-nao-index-station-based.

*Author contributions.* WMK designed the study, performed the principal analysis, and drafted the manuscript. SJGR provided the analysis
of the precipitation cycles and feedback on the methodology and results. IRS provided critical feedback throughout the analysis and on the results. SJGR and IRS contributed to the interpretation of the result and writing the manuscript. DK provided the CLM-TRENDY simulation and details about it.

*Competing interests.* The authors declare that they have no conflict of interest.

*Acknowledgements.* We thank all the research groups that produced the datasets used in this study and for making their output publicly
available. We acknowledge the Copernicus program for the ERA5 and ERA5-Land data (Hersbach et al., 2020; Muñoz-Sabater et al., 2021) available in Copernicus Climate Change Service Climate Data Store, the NASA/NOAA Global Land Data Assimilation System for the Noah and Catchment Land Surface Model datasets (Rodell et al., 2004), https://www.gleam.eu/ for GLEAM v3 (produced by Dr. Akash Koppa and validated by Dr. Petra Hulsman, Martens et al., 2017), ECA&D for E-OBS (Klein Tank et al., 2002; Klok and Klein Tank, 2009; Cornes



et al., 2018), O and Orth (2021) and Max Plank Institute for Biogeochemistry for SoMo.ml dataset, and the Climate Data guide project
at NCAR for the NAO index (Schneider et al., 2013; Hurrell et al., 2023), and the Global Carbon Project (Friedlingstein et al., 2022) for
the CLM-TRENDY simulation. WMK thanks Vít Svoboda (JILA, CU Boulder) for his comments on plotting and the drought estimation
function. WMK acknowledges funding from the Swiss National Science Foundation (grant number P500PN_206653). This work is supported
by the NSF National Center for Atmospheric Research, which is a major facility sponsored by the National Science Foundation under the
Cooperative Agreement 1852977.



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
