# Peer review of "Temporal Characteristics and Atmospheric Drivers of Onsets and Terminations of Soil Moisture Droughts in Europe"

_EGUsphere, 2024_

## Author Comment (AC1)

**Response to the reviewer 1**

We would like to thank the reviewer for his/her thorough comments and sincerely appreciate the time and effort the reviewer dedicated to reviewing our manuscript. Here, we provide our responses including the plans to address the reviewer's comments in the next revised phase.

Our responses are in blue font, and the line numbers are within [the brackets].

**General remarks**

*The manuscript is clearly structured and its topic fits to the scope of NHESS. However, there are some issues which need to be addressed by the authors before the manuscript is ready for publication.*

**Major Comments**

1. *The manuscript describes the different factors influencing onsets and terminations of droughts (precipitation and evapotranspiration) and how they differ between the datasets. However, there is almost no discussion of how exactly these differences in the precipitation and evapotranspiration lead to the differences in drought onsets and terminations in the datasets. I think a figure analogous to 7a and 7b for evapotranspiration would help to better understand the differences in the datasets. One would see how the water balances in the respective models look like in the onset and termination seasons and conclusions can be drawn about which processes are important for the duration of the onsets and terminations and which are not.*

Thanks for the reviewer's comment. We will extend the discussion on how precipitation (P) and evapotranspiration (E) explain the observed differences in the duration of onsets and terminations between the datasets.

We will take a climatological approach to this analysis. When we observe the annual cycle of E (Fig. 1 below) as the reviewer comments, GLEAM shows more reduced E during the wet period (MJJA) in CEU compared to other datasets. This period is also when drought onsets occur more frequently over the region. This reduced E may contribute to a weak decrease in P-E, not facilitating drought initiations, therefore extending onset periods. On the other hand, the discrepancies in E between the datasets are not very pronounced in MED.

[Figure]

*Fig 1. Annual cycle of evapotranspiration for each region and dataset.*

We will add these discussions in the revised manuscript, and include the cycles of E and P-E after applying the same mask over CEU and MED for precipitation in Fig 8.

2. *The description of the influence of the NAO on onsets and terminations of droughts is confusing. Positive and negative NAO phases have different effects in northern and southern Europe at different times of the year. In winter, a negative NAO means warm and humid conditions in central and northern Europe, but at the same time cold and dry conditions in southern Europe. With a negative NAO, the opposite is the case. In summer, on the other hand, a positive NAO results in warm and dry conditions in northern and central Europe, but cold and humid conditions in southern Europe. One can link a positive NAO in summer in northern Europe with anticyclonic conditions, but not in winter. I think this needs to be made clearer here. It sounds as if a positive NAO is always associated with anticyclonic conditions. This is the case for the drought onsets in summer in CEU, but does not apply the other way around. Since the NAO is particularly important in winter, a positive NAO is generally associated with strong westerlies. This potential confusion in the manuscript should be avoided, by clearly indicating which region, which season and which NAO phase is meant.*

We agree with the reviewer's comment that the difference in the seasonal impacts of NAO on the hydroclimate of these two regions was not addressed well. Note that the correlation maps in Fig. 12.a (shown below, Fig. 2) show that the correlations between NAO and soil moisture become more negative (a positive NAO inducing negative soil moisture anomalies) in MAM and JJA over CEU (in DJF and MAM over MED), which is also the point that the reviewer mentioned.

[Figure]

*Fig 2. Pearson correlation between the NAO index and monthly soil moisture anomalies. Dotted regions indicate where the correlation coefficients are statistically significant at a 95% confidence level.*

We will include clear details on these different seasonal influences of NAO on soil moisture and also extend Fig. 12.c by including mean NAO during the warm season for drought phases (see below, Fig 3) in the revised version.

[Figure]

*Fig 3. Mean NAO index value during each drought phase, also separated into cold and warm seasons for CEU (upper panel) and MED (lower panel).*

3. *The Discussion section is not really a discussion. It is just a summary of the results and some conclusions are drawn. I think that this manuscript needs a more detailed discussion of the results, setting the results of this study in context to the studies mentioned in the introduction. Where do the results agree with other studies, what is new?*

Thanks for the point. We will extend the discussion more and relate our work to the studies included in the introduction, in the revised manuscript.

Also, we would like to inform the reviewer that the results on CLM-TRENDY (from Figs. 2 to 5) will be corrected in the revised manuscript, as we found a mistake in the calculation of drought phases with this dataset. However, this is not going to change the general result and the conclusion of the manuscript.

**Minor Comments**

*Line 101-103: This sentence is almost exactly repeated at the beginning of chapter 2. Delete one of them.*

We will modify the sentence in Chapter 2 to: "The study regions are central (CEU) and Mediterranean Europe (MED), and the analyzed period is 1980-2020."

*Line 469-470: Why is influence of NAO on soil moisture droughts low? There is a strong positive correlation. Positive NAO, positive soil moisture anomaly. Isn't this the reason why onsets occur rarely during DJF. So NAO has a strong impact on the onset development in preventing it.*

We would like to express in that sentence that with positive correlations, a positive NAO does not induce negative soil moisture, therefore, no drought or onset as the reviewer comments. However, positive correlations indicate that a negative NAO would induce

negative soil moisture, therefore, the term "low" we used there might not fit in the context. In the revised manuscript, we will remove "low" and elaborate on the text more relating to the onset and termination periods.

*Line 482: positive NAO = weakening of westerlies -> In CEU in summer, should be mentioned.*

As we responded in comment 3, we will add more details on the different seasonal influences of NAO in the revised manuscript.

---

## Author Comment (AC2)

**Response to the reviewer 2**

We would like to thank the reviewer for his/her comments and appreciate the time he/she had dedicated to reviewing our manuscript. Here, we provide our responses to the reviewer's comments, in blue font.

*The investigation of drought periods, their beginning, end and duration is interesting. Dividing the study area into several sections for analysis is a good idea. However, several improvements are needed in each section of the manuscript. Therefore, I reject the manuscript at this stage.*

*1) The introduction lacks a thorough review of the literature on droughts, their processes and the relationship with the NAO in Europe. Important publications for Europe are also not mentioned. The author mentions droughts in India that are related to ENSO. But that is a completely different area with a different climate. It does not fall within the scope here.*

Thanks for your comment. We focused more on introducing past studies on drought onsets and terminations (O&T) regardless of the region, emphasizing how these studies relate drought O&T to large-scale circulation patterns. Few studies specifically investigated on soil moisture drought O&T, especially in Europe, which is the main research topic of our study. Therefore, we disagree with the reviewer's comment that the scope of some papers is off-topic, as we found it necessary to include diverse past research on drought O&T. However, we agree with the reviewer that more discussion of drought studies in Europe needs to be included. We can correct this problem in the revision phase.

*2) In the methods, I find no reference to how the data sets with different grid sizes are made comparable. This is an important point. The difference in results could be due to this very difference.*

Since all five datasets with different horizontal grid sizes show similar timing of O&T when regionally averaged (Fig. 6 in the manuscript), it is unlikely that the difference in the grid size contributes to the observed difference in the duration of O&T on a regional scale. Therefore, these differences are likely due to the models' internal physics related to soil moisture. However, when the duration is compared between the individual grid points, the difference is more visible as shown in Fig. 2 and also in Figs.4 and 5 for the occurrence timing, indicating that the grid sizes contribute to the differences on a local scale.

*3) The relationship between droughts and the NAO is not very strong. The start and end of a drought could also be more of a local phenomenon.*

Thanks for your comment. In our analysis, as the datasets show very similar onset timing corresponding to the periods with more frequent positive NAO (Figs. 6 and 13), there could be a large-scale influence of NAO on drought generation, especially over western Mediterranean regions. While it is true that the start and end of droughts can be localized phenomena, we attempt to examine O&T on a regional scale for each of the subregions in Fig. 1. We will clarify these points in the next phase.

*4) A U-test to complement the t-test might be more appropriate.*

We will consider it for the next phase.

*5) The results section needs to be restructured to improve readability.*

We will adjust that in the revision phase.

*6) The discussion section lacks a thorough discussion of the underlying processes based on previous findings. Also, convincing arguments for the link with the NAO are missing.*

As reviewer 1 also commented, we will add a separate discussion section and include more quantitative details and extensive discussion in the revised version.

*7) Many sentences are too general and lack a quantitative measure.*

As mentioned in our response 6, we will correct this in the next phase.

*8) The English language is in great need of improvement.*

We will adjust this problem in the next phase.